

# Identifying conservation priorities and management strategies based on ecosystem services to improve urban sustainability in Harbin, China

Yi Qu[1,2,*] and Ming Lu[1,*]

[1] Heilongjiang Cold Region Urban-Rural Human Settlements Science Key Laboratory, School of Architecture, Harbin Institute of Technology, Harbin, China
[2] National and Local Joint Laboratory of Wetland and Ecological Conservation, Institute of Natural Resources and Ecology, Heilongjiang Academy of Sciences, Harbin, China
[*] These authors contributed equally to this work.

## ABSTRACT

Rapid urbanization and agricultural development has resulted in the degradation of ecosystems, while also negatively impacting ecosystem services (ES) and urban sustainability. Identifying conservation priorities for ES and applying reasonable management strategies have been found to be effective methods for mitigating this phenomenon. The purpose of this study is to propose a comprehensive framework for identifying ES conservation priorities and associated management strategies for these planning areas. First, we incorporated 10 ES indicators within a systematic conservation planning (SCP) methodology in order to identify ES conservation priorities with high irreplaceability values based on conservation target goals associated with the potential distribution of ES indicators. Next, we assessed the efficiency of the ES conservation priorities for meeting the designated conservation target goals. Finally, ES conservation priorities were clustered into groups using a K-means clustering analysis in an effort to identify the dominant ES per location before formulating management strategies. We effectively identified 12 ES priorities to best represent conservation target goals for the ES indicators. These 12 priorities had a total areal coverage of 13,364 km$^2$ representing 25.16% of the study area. The 12 priorities were further clustered into five significantly different groups ($p$-values between groups $< 0.05$), which helped to refine management strategies formulated to best enhance ES across the study area. The proposed method allows conservation and management plans to easily adapt to a wide variety of quantitative ES target goals within urban and agricultural areas, thereby preventing urban and agriculture sprawl and guiding sustainable urban development.

# INTRODUCTION

Rapid urbanization and agricultural development has limited the ability of many ecosystem services (ES) in urban areas, and thus pose a threat to environmental

Corresponding author
Ming Lu, hitlm1969@hit.edu.cn, luming_hit@163.com

sustainability (*Goddard, Dougill & Benton, 2010*; *He et al., 2014*; *Kamal, Huang & Myint, 2015*). Identifying conservation priorities is essential for balancing the urgent need to protect degrading environments with the limited time, funding, and staff resources available for protection and management. Past conservation planning systems were frequently developed with biodiversity as the key criteria for delineating priority areas (*Cowling et al., 2003*; *Samways, 2007*; *Lee, Chon & Ahn, 2014*) and used a range of systematic approaches to facilitate this priority identification (*Pressey, Johnson & Wilson, 1994*; *Possingham, Ball & Andelman, 2006*; *Groves, 2003*). Although these methods have been valuable for biodiversity protection, they may overlook locations with other important ES functions due to the spatial discordance between biodiversity hotspots and ES distributions (*Chan et al., 2006*; *Yang, Zhiyun & Weihua, 2016*; *Ricketts et al., 2016*; *Morelli et al., 2017*). Consequently, there is a clear need for a method that spatially prioritizes other ES using appropriate site-selection algrithm, and then formulates management strategies based on these prioritized ES.

ES are defined as benefits that humans obtain from ecosystems and include provisioning services (e.g., food, raw material, and fresh water), supporting services (e.g., water conservation and soil conservation), regulating services (e.g., climate and flood regulation), and cultural services (e.g., landscape aesthetic and recreation) (*Reid, 2005*; *Lamarque et al., 2011*). ES are therefore seen as bridges linking natural resources and human ecological requirements (*Liu & Zhao, 2015*; *Kremer et al., 2016*). Despite some controversies over the concept of ES and its applications (*Schröter et al., 2014*; *Morelli & Møller, 2015*; *Neuteleers & Engelen, 2015*), ES are still one of the best criteria available to conduct conservation prioritization for urban areas, particularly as ES conservation represents a win-win approach compared to the traditional win-lose framework of biodiversity conservation (*Gross, 2006*). Quantifying and integrating ES that are linked to human ecosystem requirements become an important research direction, followed by the formulation of management strategies for these ES (*Liquete et al., 2015*).

Systematic Conservation Planning (SCP) is a widely used and effective method for designing conservation systems and ecological networks (*Margules & Pessey, 2000*), and thus has great potential for ES conservation and management. SCP generally involves identification of the best sites based on explicit quantitative target goals for conservation and management activities within a planning area. Irreplaceability is a key concept in SCP. It is a measure assigned to an area which reflects the importance of that area, in the context of the study region, for the achievement of the regional conservation targets. Irreplaceability can be used as measurement for conservation and management prioritization purposes (*Pressey et al., 2005*). Target goals are defined as the area or quantity of each ES indicator that should be protected; with the efficiency at meeting target goals estimated to inform how well these targets goals are represented (*Pressey et al., 2005*). These SCP features enable planners to identify priorities with explicit guidance (*Kerley, Pressey & Cowling, 2003*; *Ma, Sun & Qu, 2016*; *Kukkala & Moilanen, 2017*). Integrating ES indicators into SCP can provide a means to select the best sites for conservation and management actions that help managers meet target goals of urban sustainability. Many software products have been

developed for simplifying the site selection task. C-Plan is one of the useful planning tools. A C-Plan database can be linked to GIS interface to provide visual guidance in the planning process.

Protecting ecosystems and species are main aims of biodiversity conservation and management (*Samways, 2007*; *Lee, Chon & Ahn, 2014*). If the ecosystem conservation focus is to shift from biodiversity to ES within urban sustainable development practices, management strategies will need to move from the ecological resource itself to the functional ES products. In this management system, it will be necessary to identify the dominant ES within all possible ES conservation priorities, and to formulate more targeted management strategies for these dominant ES. The K-means clustering algorithm was effective in grouping ES priorities and identifying dominant ES in each group through the percent areal coverage of ES in the conservation priorities. This method will greatly facilitate planners in identifying dominant ES within their management priority areas, and consequently in formulating ES-targeted strategies for both conservation and sustainable development.

## MATERIALS AND METHODS

### Study area

The study area incorporates the entire city of Harbin and the surrounding rural areas within Heilongjiang Province in northeast China (125°42′E to 130°10′E and 44°04′N to 46°40′N). The study area is an administrative unit covering a total area of 53,119 km$^2$. The total population of the study area is approximately 10 million, with a human density of 200 people per km$^2$. The maximum human density is 2,694 people per km$^2$ within the Harbin Municipal District, while the minimum density is 37 people per km$^2$ in Tonghe County. The climate of the area ranges from cold temperate in the north to warm temperate in the south. Extensive forests dominate the eastern part of the study area and an east-to-west river is located in the middle north of the city. Vast urban and agricultural areas are present in the western part of the site. Excessive development of urban and agricultural zones has destroyed landscape integrity in the study area, resulting in environmental deterioration and reduced ES capacity, cascading into further ecological problems. To sustain a healthy ecosystem, the Harbin government has pursued eco-city planning within the framework of national and provincial eco-environmental development plans (*Liu et al., 2002*). The overall goal of the eco-city planning is to ensure regional sustainability, which requires further work on identifying and managing ES priority areas.

### Data sources

Three categories of data were used in this study: thematic maps, basic geographic information data, and statistical data. The thematic 2015 land use map and the 2001 vegetation map were provided by the Data Center for Resources and Environmental Sciences, Chinese Academy of Sciences (RESDC). Although the vegetation map is relatively old, it was nonetheless the most recent large-scale digital vegetation map available to us. We used the land use map to update the vegetation map based on agricultural and built-up areas, where discordant vegetation types between maps were revised according to high proportional representation of surrounding vegetation types. The soil map was provided

**Table 1  Ecological problems and ES requirements of the planning area.**

| Ecological problems | ES requirements | Corresponding ES indicators |
| --- | --- | --- |
| Water loss, soil erosion and sandification of marshland | Water conservation, soil conservation and conservation of marshland | Water conservation, soil conservation and disturbance prevention |
| Biodiversity/habitat loss | Habitats conservation of diverse species | Habitat provision |
| Flood disasters | Enhance the ability of flood regulating | Disturbance prevention |
| Deterioration of air quality | Enhance the ability of air regulating | Air and climate regulating |
| Shortage of water resources | Water source conservation | Water supply |
| Unreasonable use of forests | Scientific management of forests | Raw material supply |
| Threats to food security from rapid urbanization and unlimited sprawling of farmlands | Providing food on the basis that don't encroach important natural resources and are not encroached by urbanization | Food supply |
| Underutilization of recreation resources | Providing recreation areas for human-beings | Recreation and landscape aesthetics |

by the Environmental and Ecological Science Data Center for West China, National Natural Science Foundation of China (*Nachtergaele, Velthuizen & Verelst, 2008*). The digital elevation model (DEM) image was derived from topographic data (1:50,000) using the ArcGIS spatial analysis module (*ESRI, 2007*). The Normalized Difference Vegetation Index (NDVI) was obtained from the MODIS Satellite—NDVI product provided by NASA-USGS. Other geographic information data, including administrative zoning, river locations, and road maps were obtained from the National Geomatics Center of China. Statistical data, including rainfall and number of natural disasters, were obtained from the yearbooks of each county in the study area.

## Selecting ES indicators

Ten ES indicators in four categories were used to build the ES index system according to the ES framework of the Millennium Ecosystem Assessment (MEA) (*Reid, 2005*) and were selected using expert consultation and literature review (*Costanaza et al., 1997*; *De Groot, Wilson & Boumans, 2002*; *Reid, 2005*; *Lv et al., 2015*). The ten ES indicators are as follows: provisioning services defined as food supply, raw material supply, and water supply; regulating services defined as air and climate regulators, and disturbance prevention; supporting services defined as soil conservation, water conservation, and biodiversity maintenance; and cultural services defined as landscape aesthetics and recreational opportunities (*Costanaza et al., 1997*; *De Groot, Wilson & Boumans, 2002*; *Reid, 2005*). The ES indicator selection process also involved an analysis of locally relevant ecological problems (Table 1) and an assessment of the spatial quantifiability of the chosen indicators.

## Mapping ES indicators

To obtain the spatial distribution map of each ES indicator, we have to spaitally quantify these indicators. We found that most of the selected ES indicators (air and climate

regulating, disturbance prevention, water conservation, soil conservation, habitat provision, and recreation) could be spatially quantified using the primary geospatial datasets via the GIS-based models. Therefore we used existing GIS-based models as well as information that have been published in literatures to spatialize each of the ES indicator. Expert review workshops were necessary to identify models needed for quantifying these ES indicators. Models that are published in existing literatures and present detailed information about related determinants and their weights for building the models were selected. For ES indicators (food supply, water supply, raw material supply, and landscape aesthetic) where models were not readily available, they were instead spatially quantified based on an equivalent value generated from one or more of physical volume calculations, remote sensing images, net primary productivity (NPP), biomass simulation, or expert experience (*Xie et al., 2015*). All selected ES indicators, their spatial quantification processes, and model details are listed in Table 2.

Once the ES indicators were spatialized, they were then normalized and classified into different ranks. Normalization was accomplished using a Min-Max normalization method (*Gao, 2008*), to eliminate the influence of dimension so that data have same caliber. Six ranks per indicator were then classified via the Natural Breaks (Jenks) method that minimizes the average deviation of each class from the class mean, while simultaneously maximizing the deviation of each class from the means of other classes (*Jenks, 1967*; *ESRI, 2007*). The two highest ranks per indicator were used in our analysis to represent the spatial distribution of that indicator, thus focusing our analysis on areas with highest capabilities of providing the associated ES.

## Identifying ES conservation priorities

We used C-Plan (*Pressey et al., 2005*), one of the SCP software products and useful tool for simplifying the site/planning unit selection task, to integrate the spatialized ES indicators with their conservation target goals and to calculate an irreplaceability index in each planning unit. Details of setting planning units, setting targets goals, calculating irreplaceability and selecting priority planning units will be introduced in the following.

## Planning units

Our study area was divided into a total of 54,029 planning units, with each unit consisting of a 1 km × 1 km grid. The 1 km$^2$ grid size was selected to provide sufficient detail on ES, while not overwhelming the maximum size requirement for urban planning or the processing capabilities of C-Plan. This size can help to differentiate ecological functions in one ecosystem while at the same time providing guiding conservation and management information for outskirts of towns within the city scope. The planning unit layer was produced by the Create Fishnet and Feature to Polygon modules in ArcGIS 9.3 (*ESRI, 2007*).

## Conservation target goals

The conservation target goal for each ES indicator was defined as the explicit area of each mapped ES indicator that should be protected to guarantee maintenance of ES within the study area. The conservation target goal forms a key input into the C-Plan software used to select priority planning units in subsequent analyses. The ideal conservation target goal

**Table 2  Selected ES indicators and the process for spatial quantification.**

| ES indicators | Delineations | Spatialization methods | Involved layers | Data processing |
|---|---|---|---|---|
| *Provisioning services* | | | | |
| Food supply (*Reid, 2005*) | Ability to provide food | Assigning method based on equivalents value of food supply service | Food supply layer (Vector) | Equivalents value of food supply service supplied by per unit of different ecosystems, which is a result from literature (*Xie et al., 2015*), were assigned to the vegetation map |
| Raw material supply (*Costanaza et al., 1997*) | Ability to provide raw material | Assigning method based on equivalents value of raw material supply service | Raw material supply layer (Vector) | Equivalents value of raw material supply service supplied by per unit of different ecosystems, which is a result from literature (*Xie et al., 2015*), were assigned to the vegetation map |
| Water supply (*Costanaza et al., 1997*) | Ability to provide water | Assigning method based on equivalents value of water supply service. | Water supply layer (Vector) | Equivalents value of water supply service supplied by per unit of different ecosystems, which is a result from literature (*Xie et al., 2015*), were assigned to the vegetation map. |
| *Regulating services* | | | | |
| Air and climate regulating (*Costanaza et al., 1997*; *Groot, Wilson & Boumans, 2002*) | Regulating ability and location-criticality (*Zou, 2007*) | Multi-layer evaluation to integrate vegetation density and purification ability. The weighs were developed by AHP method | Vegetation density layer (Grid) | Represented by Normalized Difference Vegetation Index (NDVI), which was derived from MODIS/NDVI product. |
| | | | Purification ability layer (Vector) | Scores were assigned to vegetation map based on purification ability of different ecosystems in existing literature (*Zou, 2007*). |
| | | | Sensitivity layer (Grid) | Generated from suitability analysis using elevation and slope produced by digital elevation model (DEM) |

**Table 2** (*continued*)

| ES indicators | Delineations | Spatialization methods | Involved layers | Data processing |
|---|---|---|---|---|
| Disturbance prevention (*Costanaza et al., 1997*; *Groot, Wilson & Boumans, 2002*) | Disaster regulating ability and location-criticality (*Jiang, 2002*) | Multi-layer evaluation to integrate sensitivity of environment, frequency of hazard, vulnerability of affected object. Weights of the three factors were assigned based on existing literature (*Jiang, 2002*). | Frequency layer (Grid) | Interpolated from disaster frequency (rainstorm and flood) of each county using Kriging interpolation method in ArcGIS. |
|  |  |  | Vulnerability layer (Grid) | Developed by integrating population density of each county and economic density. |
| *Supporting services* |  |  |  |  |
| Soil conservation (*Groot, Wilson & Boumans, 2002*) | Soil conservation capacity ($A_c$) (*Jiang et al., 2015*) | Multi-layer evaluation based on the Revised Universal Soil Loss Equation (RUSLE) model: $A_c = R \times K \times L \times S \times (1 \times C \times P)$ (*Dai et al., 2013*; *Jiang et al., 2015*) R represents rainfall erosivity, generated according to monthly rainfall and annual rainfall (*Arnoldus, 1980*); K represents soil erodibility, constant values for different soil types provided by *Elswaify, Dangler & Armstrong (1982)*; S and L represent slope and slope length respectively. C represents the effects of different land cover types, constant values for different land covers provided by *Wischmeier (1962)* and revised by *Jiang et al. (2015)* P represents the effects of soil and water conservation measures, constant values for different land covers (*Jiang et al., 2015*) | R layer (Grid) | Produced by Raster Calculation based on monthly and annual rainfall map, which were interpolated by rainfall data of each county using Kriging interpolation method in ArcGIS |
|  |  |  | K layer (Grid) | Developed by assigning K values to soil map (*Nachtergaele, Velthuizen & Verelst, 2008*) |
|  |  |  | S layer (Grid) L layer (Grid) | Generated by DEM in ArcGIS (*Jiang et al., 2015*) |
|  |  |  | C layer (Grid) | Developed by assigning C values to vegetation map |
|  |  |  | P layer (Grid) | Developed by assigning P values to land use map. |
| Water conservation (*Lv et al., 2015*) | Water conservation capacity (*Wang & Tang, 2008*) | Suitability analysis to identify the most suitable areas for water conservation by integrating drainage area, landform, vegetation, and precipitation layers | Vegetation layer (Grid) | Developed by assigning unified scores by expert to vegetation map |
|  |  |  | Drainage layer (Grid) | Developed by assigning scores by expert to drainage map derived from DEM |
|  |  |  | Landform layer (Grid) | Developed by assigning scores by expert to DEM |
|  |  |  | Precipitation layer (Grid) | Developed by assigning scores by expert to precipitation map interpolated from rainfall data of each county |

Qu and Lu (2018), PeerJ, DOI 10.7717/peerj.4597

**Table 2** (*continued*)

| ES indicators | Delineations | Spatialization methods | Involved layers | Data processing |
|---|---|---|---|---|
| Habitat provision (*Costanaza et al., 1997*; *Groot, Wilson & Boumans, 2002*) | Habitat diversity/bio-diversity (*Margules & Pessey, 2000*) | Multi-layer evaluation to identify the areas with highest biodiversity conservation value by integrating potential distributions of indicator species (endangered or nationally protected) (*Cowling et al., 2003*). | Potential distribution of each species (Vector) | Predicted by suitability analysis based on distribution records (county as unit) in literature (*Wang & Xie, 2009*) and preferred habitats. All layers were produced and overlaid by spatial analyses in ArcGIS to represent the diversity of species. |
| *Cultural services* | | | | |
| Landscape aesthetics (*Groot, Wilson & Boumans, 2002*; *Reid, 2005*) | Value of landscape aesthetics | Assigning method based on equivalents value of landscape aesthetics service | Landscape aesthetics layer (Vector) | Equivalents value of landscape aesthetics service supplied by per unit of different ecosystems, which is a result from literature (*Xie et al., 2015*), were assigned to the vegetation map |
| | | | Landform layer (Grid) | Developed by assigning scores by expert to DEM |
| | | | Vegetation layer (Vector) | Developed by assigning scores by expert to vegetation map |
| Recreation (*Costanaza et al., 1997*; *Groot, Wilson & Boumans, 2002*; *Reid, 2005*) | Recreation suitability (*Zhang, 2008*) | Suitability analysis to identify the most suitable areas for recreation by integrating landform, vegetation, experiential value, and accessibility layers (*Zhang, 2008*). The weighs were developed by AHP method | Experience value (Grid) | Developed based on biodiversity conservation value ranks assessed in habitat provision service. |
| | | | Accessibility layer (Grid) | Developed by Straight Line Distance in ArcGIS using residence and road as source layers |

Qu and Lu (2018), PeerJ, DOI 10.7717/peerj.4597

per ES indicator is defined as the minimum amount of space required for each indicator to maintain urban sustainability. Previous publications on sustainability have suggested that this minimum target goal be set at 10%–12% of a geographical area (*Miller, 1984*), yet there has been some criticism of the science behind these relatively low threshold values (*Svancara et al., 2005*). Therefore, *Svancara et al. (2005)* suggest greater target goals based on average values for evidence-based conservation targets and average values reported from conservation assessments. In our research, we adopt the assessment-based thresholds that using potential extent of representation not total study area as the feature of interest and set our conservation targets based on area of mapped ES indicators and requirements of urban sustainable development. If the area of an ES indicator accounts for less than 10% of the total study area, the conservation target goal is set at 100% of the areal extent for this ES indicator; if the area of an ES indicator accounts for 10%–20% of the total study area, we set the conservation target at 90% of the areal extent for this ES indicator; if the area of an ES indicator accounts for 20%–50% of the total study area, we set the conservation target at 80% of the areal extent for this ES indicator; and if the area of an ES indicator accounts for more than 50% of the total study area, we set the target at 40% of the areal extent for this ES indicator.

## Irreplaceability calculation

We used irreplaceability as a measure of the importance of a particular planning unit for achieving the conservation target goals for each ES indicator (*Margules & Pessey, 2000*). Irreplaceability can be calculated by dividing the number of representative combinations that include one planning unit but would no longer be representative if this unit were removed by the total number of representative combinations (*Pressey et al., 2005*). Planning units with high irreplaceability will be given high priority for selection as natural reserves or as areas with special management, and thus can be used to guide urban planning processes (*Ferrier, Pressey & Barrett, 2000*). The irreplaceability index was calculated in the C-Plan software.

## Identification of ES conservation priorities

Continuous values of irreplaceability can be divided into different ranks (very high, high, medium, low and very low) via the Natural Breaks (Jenks) method. A high irreplaceability rank indicates planning units with high conservation value and ES functions. Planning units with very high and high irreplaceability ranks were identified as ES priority areas. To facilitate further conservation and management, ES priority boundaries were then re-defined according to town boundaries and land-use borders. Twelve ES priorities that were composed of high irreplaceability planning units were identified within the study area.

## Grouping ES priorities and formulating management strategies

We used a K-means clustering algorithm to divide the 12 ES priorities into smaller set of groups based on dominant ES indicators. First, the proportional area of each ES indicator for each priority was calculated to obtain the dominance degree of the indicator. A matrix of proportional areas for each ES indicator in each ES priority was created. This matrix was used as input for the K-mean clustering analysis. The K-means clustering analysis classified

the ES priorities based on the dominant ES within each ES priority. Iterative application of the K-means clustering identified five clusters as the most realistic representation of the data. Therefore, five groups of ES priorities were classified, and management strategies for the five groups were formulated based on the common dominant ES in each group. For groups dominated by supporting services, a protective management strategy was suggested. Management for groups that were dominated by regulating ES services should require increased green infrastructure. If the dominant ES in a group was a provisioning service, the management strategy should involve adaptive utilization. If the dominant ES was cultural services, the preferential strategy should be recreational adaptations. Finally, if there were more than one types of services as dominant ES at a priority, the management strategy should combine the relevant above-mentioned preferential strategies.

## RESULTS

### ES conservation and management priorities

The spatial patterns of irreplaceability across the study area are shown in Fig. 1. The irreplaceability values were classified into five ranks: very high, high, medium, low, and very low (Table 3). We identified areas with very high and high irreplaceability values as ES priorities. 13,364 km$^2$ of land was classified as having a very high or high irreplaceability (accounting for 25.16% of the study area) and were mainly located along the Songhua River (the main water body of Harbin city), in the forested areas at the northern and southern riverbanks, and in a large strip of forest along the southeast margin of the city. The river area was subdivided into four separate ES priorities given different dominant ES functions. 12,401 km$^2$ of land was classified as having a medium irreplaceability (accounting for 23.35% of the total study area) and were mainly located adjacent to locations with very high irreplaceability values. Low and very low irreplaceability areas cover 27,353 km$^2$ of land (accounting for 51.50% of the study area).

The ES priority boundaries were then re-defined by town boundaries and land use borders to generate more manageable management units. The revised ES conservation priorities covered 14,765 km$^2$ (accounting for 27.80% of the study area), which is 1,401 km$^2$ larger than unrevised results.

### Target goals achievement

Efficiencies for meeting conservation target goals of the revised ES priorities are listed in Table 4. Average efficiencies were high, indicating that if the planned ES priorities were properly preserved or managed, 77.51% of the conservation target goals would also be met. In fact, five of the ten ES indicators exceeded 80% efficiency for conservation target goals. Only water conservation was over-represented with an efficiency above 100%. All other efficiencies exceeded 60% except for food supply, which only achieved about 25% of the conservation target goal in our analysis. Ideal conditions dictate that all conservation target goals reach an efficiency of 100%, but this situation would result in too many scattered sites making real-world conservation and management difficult. Conversely, the revised ES priorities generated in this study emphasize core areas of concern, with their surrounding areas included as buffer zones for preventing further external disturbances. Inclusion

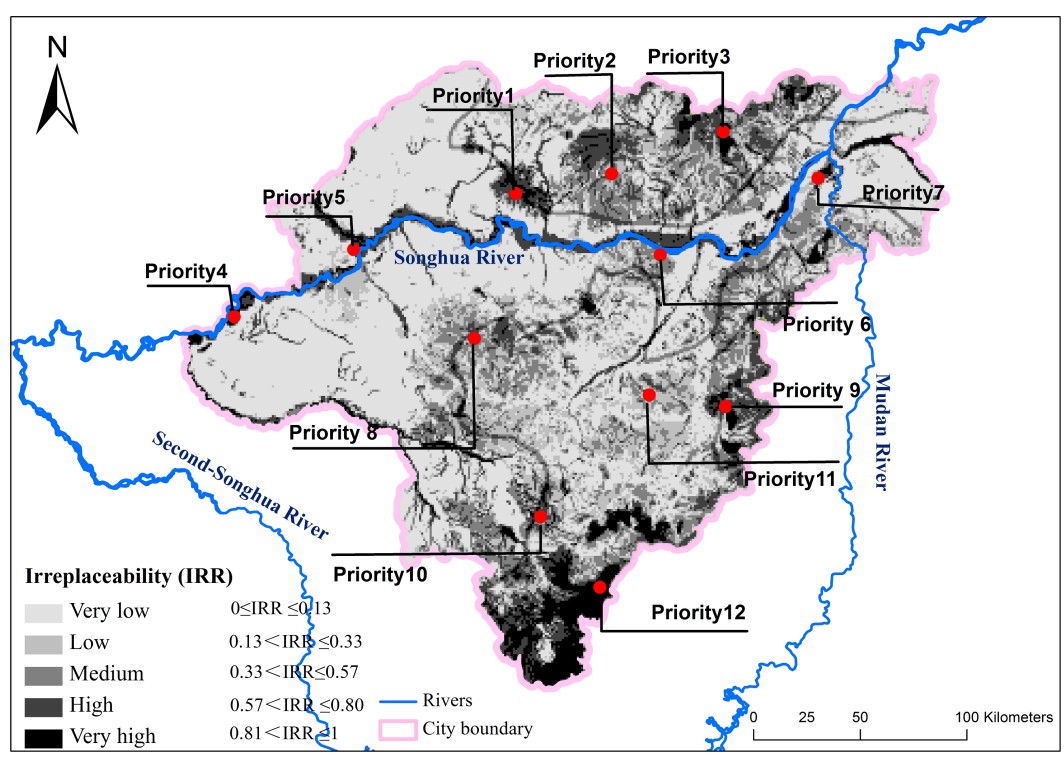

**Figure 1** The spatial patterns of irreplaceability.

**Table 3** Characteristics of different levels of IRR.

| IRR levels | IRR | Area (km²) | Percentage of the total area (%) | Description |
|---|---|---|---|---|
| Very low | 0.00–0.13 | 22,534 | 42.43 | Very low value for ES conservation |
| Low | 0.13–0.33 | 4,819 | 9.07 | Low value for ES conservation |
| Medium | 0.33–0.57 | 12,401 | 23.35 | Medium value for ES conservation |
| High | 0.57–0.80 | 8,238 | 15.51 | High value for ES conservation |
| Very high | 0.80–1.00 | 5,126 | 9.65 | Very high value for ES conservation |

of lands assessed as having moderate irreplaceability in ES priorities would increase the efficiency for meeting the designated conservation target goals to close to 100% for each goal.

## Management strategies guided by dominant ES in priorities

The percent aerial coverage of each of the 10 ES indicators in each priority (ES priorities 1–12) is shown in Fig. 2. Most of the ES priorities contain all ES indicators, except for ES Priority 4 (Fig. 2D), ES Priority 5 (Fig. 2E), ES Priority 6 (Fig. 2F), and ES Priority 11 (Fig. 2K). Priority 4, 5, and 6 all lack soil conservation and water conservation indicators, and priority 11 lacks water supply and landscape aesthetics. In each ES priority, the percent areal coverage of the ES indicators is not equal, indicating that the different priorities have different dominant ES functions. For instance, habitat provision, recreation, disturbance

**Table 4  Calculation of efficiency for meeting target goals of ES indicators.**

| ES indicators | | Potential target achievement (%) |
|---|---|---|
| Provisioning services | Food supply | 25.48 |
| | Raw material supply | 69.38 |
| | Water supply | 78.78 |
| Regulating services | Air and climate regulating | 66.82 |
| | Disturbance prevention | 90.26 |
| Supporting services | Water conservation | 100.11 |
| | Soil conservation | 88.63 |
| | Biodiversity maintenance | 80.39 |
| Cultural services | Landscape aesthetics | 78.78 |
| | Recreation | 96.48 |

prevention, and water conservation are the main ecosystem service indicators in ES Priority 1, while disturbance prevention, raw material supply, and air and climate regulating are the main service indicators in ES priority 2.

The mean percent areal coverage of the different ES indicators per ES priority highlights the diversity of ES. The following order was found for all 12 priorities: Priority 12 (54.45), Priority 3 (49.46), Priority 4 (49.19), Priority 6 (47.77), Priority 9 (47.17), Priority 2 (43.92), Priority 10 (43.09), Priority 1 (42.38), Priority 7 (40.57), Priority 5 (39.04), Priority 8 (38.06), and Priority 11 (37.71). The mean percent coverage of each ES indicator over the 12 ES priorities is ordered as follows: disturbance prevention (84.38), recreation (69.91), raw material supply (66.53), air and climate regulating (58.96), biodiversity maintenance (53.37), water supply (29.08), landscape aesthetic (29.08), food supply (25.81), water conservation (17.35), and soil conservation (9.55). These result show that the ES priorities within the study area have a low potential to offer water and soil conservation ES, while they have a high potential to regulate disturbances and supply raw materials for human use.

The 12 ES conservation priorities were subsequently classified into five groups through the K-means clustering analysis. ANOVA results indicated significant differences between each of these five priority groups (all $p$-values <0.05 for multiple means comparisons). Group 1 contains only ES Priority 1; group 2 contains ES Priority 5 and 6; group 3 contains ES Priority 2, 8, 10, and 11; group 4 contains ES Priority 4 and 7; and group 5 contains ES Priority 3, 9, and 12. The final results for the cluster center indicate that, compared to other groups, group 1 has the strongest capacity for water conservation; group 2 has the strongest capacity for habitat provision (biodiversity) and food supply; group 3 has the strongest capacity for air and climate regulation; group 4 has the strongest capacity for habitat provision, water supply, and landscape aesthetics; and group 5 has the strongest capacity for water and soil conservation. The management strategies developed based on the above-mentioned characteristics of each group are shown in Fig. 3. Counties surrounding ES priorities of group 1 should focus on conserving and managing water resources and species habitats to ensure water security and to maintain biodiversity;
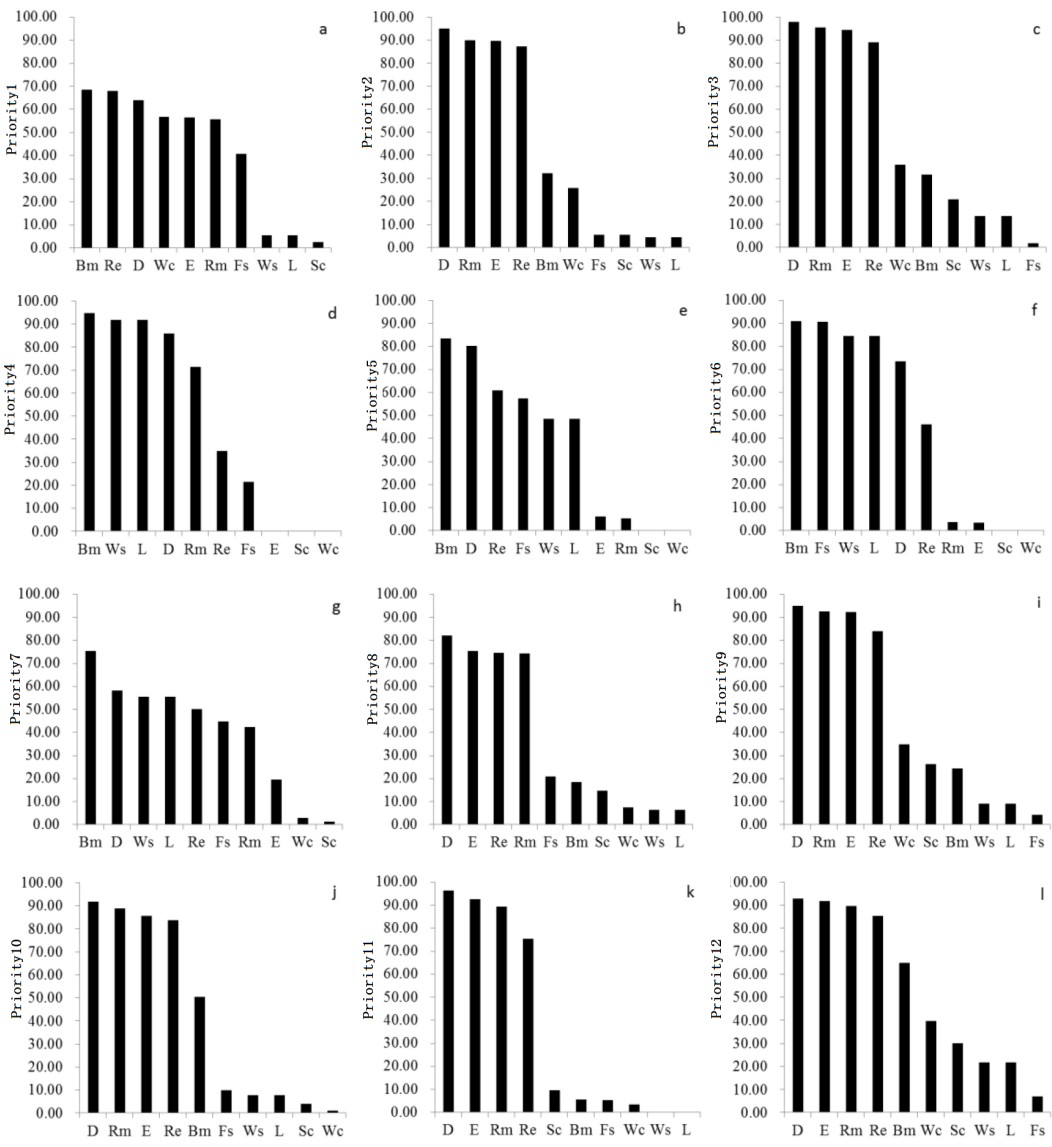

**Figure 2** **The area percentage of each ES indicator in each priority.** (A) ES indicators and their area percentages in Priority 1; (B) ES indicators and their area percentages in Priority 2; (C) ES indicators and their area percentages in Priority 3; (D) ES indicators and their area percentages in Priority 4; (E) ES indicators and their area percentages in Priority 5; (F) ES indicators and their area percentages in Priority 6; (G) ES indicators and their area percentages in Priority 7; (H) ES indicators and their area percentages in Priority 8; (I) ES indicators and their area percentages in Priority 9; (J) ES indicators and their area percentages in Priority 10; (K) ES indicators and their area percentages in Priority 11; (L) ES indicators and their area percentages in Priority 12.

counties surrounding ES priorities of group 2 should focus on conserving habitats for wetland birds and developing sustainable agricultural practices; counties surrounding ES priorities of group 3 should focus on establishing national parks that are closely related to the living quality of human beings; counties surrounding ES priorities of group 4 should focus on conserving and managing water resources and biodiversity while also exploring

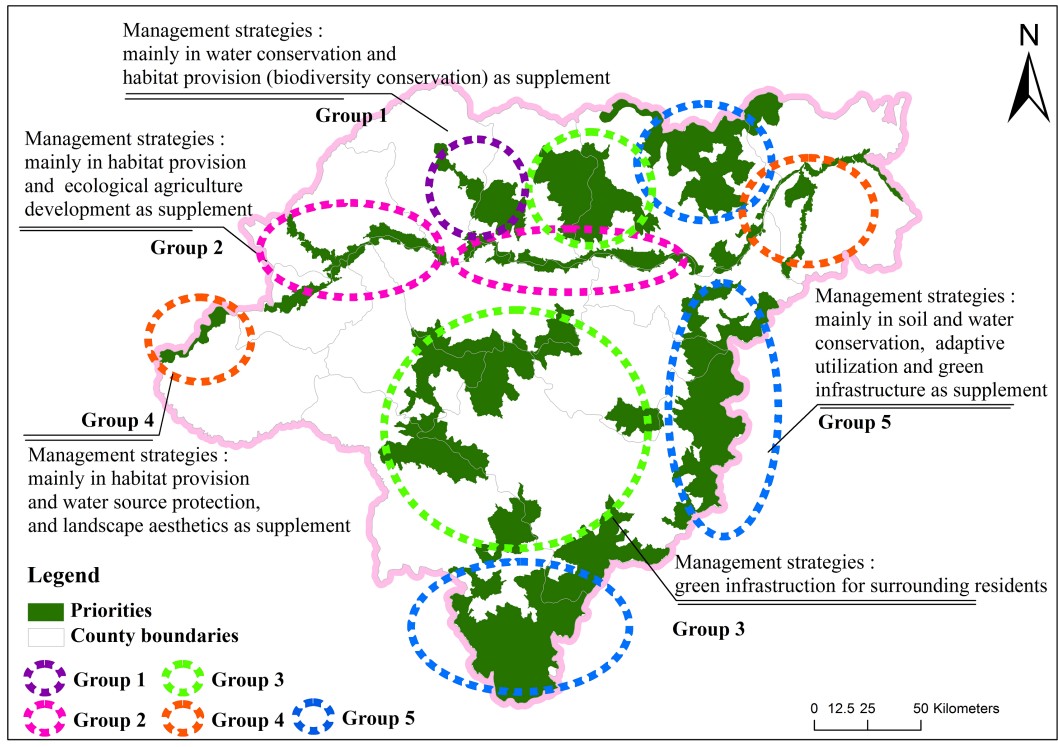

**Figure 3** **The management strategies developed based on above characteristics in each group.**

the aesthetic value of the area for tourism; and counties surrounding ES priorities of group 5 are suitable for comprehensive development activities, but are encouraged to enforce strict protection of important water and soil conservation areas, implement adaptive utilization of buffer forest for the sustainable supply of raw materials, and focus on improving the quality of life for residents through the establishment of national parks.

# DISCUSSION

The identification of conservation priority areas and management strategies based on ES is invaluable for urban sustainability. In this study, we demonstrated the effectiveness of an alternative ES-based framework for establishing conservation priority areas and management strategies using systematic conservation planning (SCP) and a K-means clustering analysis for the case study area of Harbin, China.

The SCP method has been shown to be an effective approach for designing biodiversity conservation plans (*Cowling et al., 2003*; *Samways, 2007*; *Lee, Chon & Ahn, 2014*), yet very few studies have looked specifically at ES prioritization within the SCP framework (*Samways, Bazelet & Pryke, 2010*; *Mubareka et al., 2013*). Our research sought to fill this knowledge gap by showing that multiple ES can be effectively integrated into conservation priority planning approaches. In response to the spatial discordance between biodiversity and ES functions, we incorporated biodiversity into ES system via representing biodiversity as diverse species habitats (habitat provision is an important ES function). Our SCP results

explicitly reveal the relative importance of different spatial planning units for achieving target goals of both biodiversity conservation and ES enhancement. The identification of priorities with high probabilities for accomplishing conservation target goals further illustrates the usefulness of SCP for enhancing ES as part of sustainable urban development.

Results of the K-means cluster analysis also reveal how different ES priorities can be grouped based on dominant ES indicators. Past studies have often formulated management strategies based strictly on the conservation value for different management actions (for example, high value given for strict protection, medium value for buffering interruptions, and low value for economic development), and thus little research has been done on the dominant function of different ES (*Liquete et al., 2015*). Our K-means clustering results identified five groups dominated by different ES, thus allowing us to formulate management strategies with more targeted conservation treatments. ES functions such as supporting and regulating services should have management prioritized toward protection, while functions such as provisioning and cultural services should be considered comprehensively with adaptive utilization practices including multifunctional agriculture, sustainable intensification, and conservation agriculture (*Friedrich, Kienzle & Kassam, 2009*; *Leakey, 2012*; *Dile et al., 2013*). In general, these ES prioritization methods and associated management strategies provide important guidelines for implementing conservation system with limited funds. The proposed framework is easily applicable to other areas while adjusting for regional differences in the weights used in the GIS models and parameters input in the K-means cluster analysis.

This study effectively demonstrated a new method for identifying conservation priorities and formulating management strategies using ES, yet further research is needed to fully optimize the link between ES and sustainability management. Of primary concern is developing a better method for setting conservation target goals, since these goals are critical for ES priority site identification given they are the driving factors for the irreplaceability calculations. Any change to these target goals can result in variations of irreplaceability patterns. Setting effective conservation target goals can be difficult due to varying ES requirements and the lack of well-developed guidelines. In this study, we set our conservation target goals based on the proportional area of potential distribution area of the ES indicators. However, the approach can be refined to set these goals based on scientific quantification of ES requirements, but this will require advancing our understanding of the required minimum area or quantity for each ES indicator needed to ensure the sustainability of the planning area (*Schulp, Lautenbach & Verburg, 2014*; *Rodríguez et al., 2015*). Unfortunately, the lack of basic data to quantify ES requirements hinders further research in this direction. Although there is an inherent uncertainty in our conservation target goals in this study, they do offer effective benchmarks for work within the planning process. Nonetheless, once reliable data or estimation methods become available, the conservation target goals should be revised to improve the effectiveness of our method. Socio-economic factors, such as opportunity costs and local financial support, should also be taken into account in ES conservation priority identification (*Adams, Pressey & Naidoo, 2010*; *Knight et al., 2011*), yet, monetary values for acquisitions, compensations, and transaction costs (*Keppel et al., 2012*; *Pressey et al., 2013*; *Iftekhar et al., 2016*) are typically

unavailable in our study area. Consequently, we have to fall back on cost estimations, which are complicated in ES conservation and management due to the complex relationship between ES providers and ES receptors, as well as the dynamic relationships between costs, ES conservation values, and the feasability of implementing conservation and management (*Manhães et al., 2018*). Developing both reliable and broadly applicable metrics for ES conservation and management costs will require intensive research to better identify spatial variability in costs in relation to the specific ES priority areas. The critical next step for conservation scientists will be to engage with relevant economic experts in developing more robust prioritization schemes that comprehensively consider explicit conservation target goals with realistic measures of ES conservation and management costs (*Game, Kareiva & Possingham, 2013*).

## CONCLUSIONS

In this study, we proposed a spatially explicit approach for designing a conservation and management plan for Harbin, China. This plan identifies ES priorities and formulates management strategies based on the dominant ES of clustered priority site groups. The conservation priorities are areas that have a suitable biogeographic environment for the generation of diverse ES functions. Effective conservation and management of these areas is expected to considerably conserve the provision of ES, enhance the social well-being of surrounding communities, and ensure urban sustainability. Clustered ES priority groups that share the same set of dominant ES indicators have been carefully identified, and corresponding management strategies have been formulated to guide further development in surrounding cities and towns.

In general, the approach proposed here allows for conservation and management planners to meet a wide array of ES target goals in both urban and agricultural contexts. First, it prioritizes ES functions for specific target areas to improve management systems. Second, it allows conservation and management measures to be spatially explicit in meeting conservation target goals. Third, it guides further definition of conservation planning sub-regions based on the most important ES functions provided by local natural resources. Finally, it is adaptable to other regions and can guide sustainable development in areas under threat from urban and agricultural sprawl.

### Funding

Funding for data surveys and collection came from the projects of National Natural Science Foundation of China (Project Number 41501583 and 51438005) and Youth Fund of Heilongjiang Academy of Science (CXMS2017ZR01). The funders had no role in study design, data collection and analysis, decision to publish, or preparation of the manuscript.

### Grant Disclosures

The following grant information was disclosed by the authors:

National Natural Science Foundation of China: 41501583, 51438005.
Youth Fund of Heilongjiang Academy of Science: CXMS2017ZR01.

## Competing Interests

The authors declare there are no competing interests.

## Author Contributions

- Yi Qu conceived and designed the experiments, performed the experiments, analyzed the data, contributed reagents/materials/analysis tools, prepared figures and/or tables, approved the final draft.
- Ming Lu conceived and designed the experiments, prepared figures and/or tables, authored or reviewed drafts of the paper, approved the final draft.

## Data Availability

The raw data are provided in Supplemental Information 1.

## Supplemental Information

Supplemental information for this article can be found online at http://dx.doi.org/10.7717/peerj.4597#supplemental-information.

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
