# Peer review of "Identifying conservation priorities and management strategies based on ecosystem services to improve urban sustainability in Harbin, China"

_PeerJ, doi:10.7717/peerj.4597_

## Round 0.1 · original submission · Major Revisions

I carefully read the comments by referees. They look very useful and I think you can strongly rewrite your manuscript.

Reviewer 1 ·

Basic reporting

The idea to identify areas of ES priorities in urban areas using as surrogate "very high and high irreplaceability values" it's a very interesting strategy, potentially useful in ecological management, and for the development of conservation strategies - as highlighted also by the authors. The idea of the paper is theoretically useful and sounds suitable for the targeted journal. However, from my point of view, some points and concerns need to be addressed before any consideration of the publications in the Journal.

Experimental design

The experimental design sounds adequate to address the main question raised by authors. The idea to identify areas of ES priorities in urban areas using as surrogate "very high and high irreplaceability values" could be a suitable and cost-effective for ecological planning and identification of conservation priority areas.
I suggested to authors to provide additional explanations about some points, detailed in the section "comments for authors".

Validity of the findings

The findings could be useful also in other contexts, and probably - after a careful correction of some points - the transferability of the study could be more clear.

Additional comments

Specific comments to the authors
Abstract:
"Rapid urbanization...." is causing the degradation of "ecosystems" more than degradation of "ecosystem services". I would explain: If authors focus for example on "cultural ecosystem services" the growth of urbanization cause even an improvement on this kind of ES. I suggest to modify the sentence as
"Rapid urbanization and agricultural development resulted in the degradation of ecosystems, many time affecting also the ecosystem services (ES) and posed a significant threat to urban sustainability."

Introduction
Lines 30-32: "Rapid urbanization...." is causing the degradation of "ecosystems" more than degradation of "ecosystem services" or decreasing the ability of all ES. I would explain: If authors focus for example on "cultural ecosystem services" the growth of urbanization cause even an improvement on this kind of ES. I suggest to modify the sentence as
"Rapid urbanization and agricultural development decreased the ability of many ecosystem services (ES) and thus posed a great threat to sustainability at urban or larger scale (Goddard et al. 2010; 32 He et al.2014; Kamal et al. 2015)."
Or something similar…

Lines 38-41:
"While the identified conservation priorities in the conservation systems have been valuable for biodiversity protection, these priority sites may have missed other areas with important ES functions due to the spatial discordance between biodiversity hotspots and ES distributions (Chan et al. 2006; Yang et al. 2016)."
I suggest to authors to add also this references in this sentence:
Morelli, F., Jiguet, F., Sabatier, R., Dross, C., Princé, K., Tryjanowski, P., Tichit, M., 2017. Spatial covariance between ecosystem services and biodiversity pattern at a national scale (France). Ecol. Indic. 1–13. doi:10.1016/j.ecolind.2017.04.036
Ricketts, T.H., Watson, K.B., Koh, I., Ellis, A.M., Nicholson, C.C., Posner, S., Richardson, L.L., Sonter, L.J., 2016. Disaggregating the evidence linking biodiversity and eco- system services. Nat. Commun. 7, 13106. http://dx.doi.org/10.1038/ncomms13106.

Lines 46-50: Please, verify that the most accepted definition the ES are classified into provisioning, regulating and cultural services.

Lines 51-54:
Even if I agree with authors that the ES approach could be useful especially in urban-related areas, I suggest to cite some of the main criticisms moved against the ES approach. Some examples here:
Schröter, M., van der Zanden, E.H., van Oudenhoven, A.P.E., Remme, R.P., Serna-Chavez, H.M., de Groot, R.S., Opdam, P., 2014. Ecosystem services as a contested concept: a synthesis of critique and counter-Arguments. Conserv. Lett. 7, 514–523. http://dx. doi.org/10.1111/conl.12091.
Morelli, F., Møller, A.P., 2015. Concerns about the use of ecosystem services as a tool for nature conservation: From misleading concepts to providing a “price” for nature, but not a “value.” Eur. J. Ecol. 1, 68–70. doi:10.1515/eje-2015-0009
Neuteleers, S., Engelen, B., 2015. Talking money: How market-based valuation can undermine environmental protection. Ecol. Econ. 117, 253–260. doi:10.1016/j.ecolecon.2014.06.022

Methods
Study area
Lines 92-93: Please, provide more information about the city of Harbin: population, human density. This information could be useful to represent the type of city focused, giving additional information complementary to the environmental description presented in the next paragraphs.

Lines 99-101:
"If the ecosystems were not protected in time...."
This sentence looks more like a sentence for Introduction section than for Methods. Please, re-word.

Lines 123-124:
Are authors providing a summary about the procedure followed by experts to identify and give weights to each indicator? Please, explain better.

Line 133: please, add a "space" between "of" and "54029"

Lines 136-138:
Authors need to cite some references as demonstration of this statement. Who guarantee that this spatial scale is enough to provide detailed conservation and management guidance? Please, provide some examples of studies focused in the same spatial scale.

Line 140:
Add a reference for ArcGIS 9.3

Lines 149-150:
Authors need to consider moving this sentence to AIMS section, in the Introduction section.


Line 156:
Redundant information. Please, re-word in "Methods style". Comments like "Irreplaceability is the key concept in SCP" and other similars are comments for Introduction, not for Methods.

Lines 166-167:
Can authors explain what means "were revised" in this context?

Results
Lines 189-190: The idea to identify areas of ES priorities in urban areas using as surrogate "very high and high irreplaceability values" it's a very interesting strategy.

Lines 197-198: "These areas could be viewed as...." its a sentence not adequate for Results section. Authors have to move to the Discussion section.

Lines 199-201: "This is mainly due to extensive farmlands...." Move to Discussion section.

Line 202:
Define "revised".

Line 235:
Please, introduce a space between "all" and "12".

246: Authors need to remove this sentence from here or to merge into the Methods section. In Results section authors must be limited to present with clarity ONLY results.

Lines 250-252: "Based on the ES functions and vegetation types each priority contained, we suggest management and control policies for each priority group."
This is clearly not a Results sentence. Please, move or re-word.

Line 254: Define "most powerful" in this context.

Discussion
General comment: I warmly suggest to authors to shorten the Discussion section. Currently is too long, and sometimes a little bit trivial. Please, try to focus mainly on your findings.

Reviewer 2 ·

Basic reporting

This paper has acceptable English but in some sections are not clear/professional enough to understand. It contains some literatures but the background information is not sufficient. The structure is fine and all the sections are clearly divided and raw data shared. Results and discussions are also shown but still not sufficient.
1. English needs improve and double-check, words such as "suitable ranks","excellent method""good knowledge of method", etc. should not be the way in a scientific paper.
2. Literatures are not enough to understand the background. This paper used GIS, C-plan and K cluster but no explanation about any of them. It referred "existing GIS model" many times in line 79/83/88/120 and table 2 but what model in detail? GIS contains numberless models and you can also build your own model, which one/are the models you choose and why? ArcGIS 9.3 or ArcMap?
This paper might be a continuous study based on Zuo 2007 and Xie 2015 (it referred these two papers a couple of times), but it doesn't automatically lead all the audiences to understand the model. Please remember they are separate papers. Similar questions also appear in C-plan and K cluster, the two main software/analysis methods. No background information on what are they? why use them and how did they used in detail? So the Introduction and Method parts need substantial changes to improve. Line 42-45 for example, it doesn't need to repeat the aim of this paper in introduction part several times but show the research gap and how you aim to deal with the research questions. In Section 2 Materials/Method part, what are the data sources? What expertise/literatures you referred to select ES indicators?

Experimental design

This paper studies at ecosystem services spatially and how this meet the aim of local contexts. So it meets the aims and scope of the journal. This research question defined well and meaningful. But my main concern is the way the authors presented the methods.
1. As mentioned above, not enough background information are described, particularly the main software/analysis methods. The 10 ES indicators should be explained earlier than the result part since not many people can understand immediately what the 10 indicators are. This is also relevant to understand the Fig.2 where you use abbreviation but no information about them were provided before.
2. This paper referred Miller's study but still not sufficient information was provided, such as why 10%-12%, the same question is about irresplaceability. What is it and why is it important?
In all, the method part needs substantial improvement.

Validity of the findings

The result part is fine but minor misunderstanding is that the authors used priority 1-12 in Figure 1 and then a-I in Figure 2. These should be consistent. Data is sound and controlled. My concerns is Discussion part. Discussion still more like results but should be with more leap it up to future studies, such as how this contribute to current ES, what is the weakness and how the authors plan the future studies.

Additional comments

This is an empirical paper based on big data analysis. The next step is to make the paper more clear and lift it up to further discussion and avoid replications.

Reviewer 3 ·

Basic reporting

no comments.

Experimental design

It is good in the experimental design for the present study; but some info should be added.

Validity of the findings

Most of the findings are supported by the data and analysis.

Additional comments

Comments on PeerJ #22812

Title: Identifying priority conservation sites and management strategies based on ecosystem services for urban sustainability in Harbin, China

The present study tried to propose a comprehensive framework for the identification of ES priorities and management strategies for planning areas in Harbin, a northern city in China.

I do think this paper is a good start and it could make a contribution to city management in China.

Major comments:

1. I could not find the reference list in the Reviewing PDF or WORD file.
This is my first time to review a manuscript without showing Reference List. Thus, my comments did not include any reference. However, in some cases, references would be very important for the reviewer to do better.
For example, line 111 said that literature review methods were used to select ES indicators. But I could not find related references for this.

2. In the Discussion section, you should show firstly that the main findings of your study.
Thus, I feel that lines 271-285 are good for any paper, for any city; instead, I did not know if it was good for your study area.
It was the same case for lines 286-295, and even perhaps for lines 296-310.
I found most of your discussion was not related to any of your result, and they could be used for any ES and SCP paper.

3. Similar to point 2, in the section of Abstract: lines 11-15 are good; however, after line 16, it seemed no results and conclusions.

Minor comments:

Line 5: changed to “…Heilongjiang Academy of Sciences, Harbin 150040, China”

Line 6: changed to “…Harbin 150001, China”

Lines 107-110: For the ES prioritization, 10 ES indicators were selected to represent the ES that are required by the planning area under the framework of the Millennium Ecosystem Assessment and available GIS models or previously published results.
Here, please explain more in details about “available GIS models or previously published results”.

Lines 115-116: please Ref for national research agencies.

---

## Round 0.2 · accepted · Accept

I personally have some restrictions on ES concept, but now your vision is much better presented, and I think you adopted some other view on the subject. Moreover, we need similar data from China, not only from N America and Europe - as majority studies mentioned.

I have checked your revision and believe it appropriately addresses the comments of the reviewers.

Hence, I decide to accept your manuscript.